# Critical Review on Fatty Acid-Based Food and Nutraceuticals as Supporting Therapy in Cancer

**DOI:** 10.3390/ijms23116030

**Published:** 2022-05-27

**Authors:** Carla Ferreri, Anna Sansone, Chryssostomos Chatgilialoglu, Rosaria Ferreri, Javier Amézaga, Mercedes Caro Burgos, Sara Arranz, Itziar Tueros

**Affiliations:** 1Istituto per la Sintesi Organica e la Fotoreattività, Consiglio Nazionale delle Ricerche, Via Piero Gobetti 101, 40129 Bologna, Italy; anna.sansone@isof.cnr.it (A.S.); chrys@isof.cnr.it (C.C.); 2Department of Integrated Medicine, Tuscany Reference Centre for Integrated Medicine in the Hospital Pathway, Pitigliano Hospital, ASL Sudest Toscana, 58017 Pitigliano, Italy; rosaria.ferreri1957@gmail.com; 3AZTI, Food Research, Basque Research and Technology Alliance (BRTA), Parque Tecnológico de Bizkaia, Astondo Bidea, Edificio 609, 48160 Derio, Spain; jamezaga@azti.es (J.A.); mcaro@azti.es (M.C.B.); sarranz@azti.es (S.A.); itueros@azti.es (I.T.)

**Keywords:** anticancer strategy, fatty acid signaling, membrane fatty acids, membrane lipidomics, dietary fatty acids, precision nutraceuticals

## Abstract

Fatty acids have an important place in both biological and nutritional contexts and, from a clinical point of view, they have known consequences for diseases’ onset and development, including cancer. The use of fatty acid-based food and nutraceuticals to support cancer therapy is a multidisciplinary subject, involving molecular and clinical research. Knowledge regarding polyunsaturated fatty acids essentiality/oxidizability and the role of lipogenesis-desaturase pathways for cell growth, as well as oxidative reactivity in cancer cells, are discussed, since they can drive the choice of fatty acids using their multiple roles to support antitumoral drug activity. The central role of membrane fatty acid composition is highlighted for the application of membrane lipid therapy. As fatty acids are also known as biomarkers of cancer onset and progression, the personalization of the fatty acid-based therapy is also possible, taking into account other important factors such as formulation, bioavailability and the distribution of the supplementation. A holistic approach emerges combining nutra- and pharma-strategies in an appropriate manner, to develop further knowledge and applications in cancer therapy.

## 1. Introduction

Lipids are mostly fatty acid-containing molecules, exemplified by the representative structures of triglycerides, phospholipids and cholesteryl esters, containing three, two and one fatty acid chains, respectively [1]. Fatty acids have an important place in both biological and nutritional contexts, and from a clinical point of view they attract more and more attention for their involvement in diseases’ onset and development, including cancer. Connections between the molecular and clinical aspects rely on two main aspects: the first aspect, first studied in structural biology but not limited to this field, is the role of fatty acids in compartment formation and organization, necessary to drive protein–lipid interactions and functions [2]. The hydrophobicity of the fatty acid chains is the driving force for “compartmentalization” in cells [3]. Indeed, the spontaneous organization of the fatty acid chains occurs in order to avoid contact with the predominant extra- and intra-cellular aqueous phase. The composition and roles of the different types of extra- and intra-cellular lipid organizations were increasingly studied, evidencing crucial differences between the plasma membrane, endoplasmic reticulum and organelles [4]. Membrane biophysical studies highlighted specific differences of organization due to the ratio between saturated or unsaturated fatty acid structures, with direct influence on membrane properties and lipid rafts’ formation, which have functional and clinical consequences described in several reviews [5,6,7]. The basic process of membrane formation, that generates life by the formation of cells, occurs constantly in living organisms, however, in cancer the quality and quantity of fatty acids used for such spontaneous organization assumes crucial relevance for cell growth that sustains disease proliferation.

The second connection between molecular and clinical aspects concerns the presence of fatty acids in cellular pools with their nutritional intakes, provided mostly by triglycerides (ca. 80% of the lipid dietary intakes are triglycerides) and the digestion–absorption-distribution pathways. By these pathways, cellular lipid pools are created in all of the tissues by a mix of saturated and unsaturated fatty acids, to be involved in the continuous processes of cell replication and compartment formation. It is worth recalling that the dietary intakes are particularly important for polyunsaturated fatty acids (PUFA), which cannot be formed by eukaryotic—hence human—cells, since they do not have the enzymatic arsenal to go from the endogenously formed monounsaturated omega-9 fatty acids to the polyunsaturated omega-6 and omega-3 structures [8]. Therefore, PUFA are “essential” in our diet for allowing cell formation, with established nutritional daily intake values by the main international food agencies [9], but this essentiality is also vital for cancer cells.

The molecular structures of the mono- and poly-unsaturated fatty acids (MUFA and PUFA) are connected through metabolic cascades, and Figure 1 shows the main pathways and products: the endogenous pathway of saturated fatty acids (SFA), then transformed to MUFA, starts from palmitic acid (16:0, SFA), synthesized by the enzymatic system of Fatty Acid Synthase (FAS). On the other hand, the PUFA pathways start from the intakes of linoleic acid (18:2, PUFA omega-6) and *α*-linolenic acid (18:3, PUFA omega-3) as dietary precursors, which are essential fatty acids (EFA). It is worth underlining the conversion of C16 and C18 fatty acids into C20 and C22 fatty acids, which are then further transformed into bioactive lipids (i.e., eicosanoids and docosanoids). It is logical that the original PUFA intakes determine the levels of such transformations, thus influencing the related biological effects.

The carbon chain length and the double bond number and position create chemical, physical and biophysical diversity, beside diverse biological effects, which are needed for the various structural and functional roles in cells. It is not the scope of this review to go into details that were already treated in books and reviews [11,12]. Fatty acids derived from food or supplementation are incorporated in the fatty acid cellular pools, and have the same potential for entering or influencing the biological balance with respect to the endogenously produced molecules. This fact is more and more understood, and fatty acids from the diet are increasingly considered for such effect, that renders them useful to support pharmacological interventions, and also in cancer. The applicability of fatty acids in this context is a multidisciplinary subject of biological, pharmacological and clinical research, and, in the sections below, some aspects of fatty acid-based foods and nutraceuticals in cancer therapies will be highlighted.

## 2. Essentiality of PUFA: Indications from the Membrane Composition

The essentiality of PUFA finds its highest expression in the creation of biological membranes, including cancer cell membranes, which cannot be formed without taking them from the diet. Therefore, there is an indissoluble correlation between dietary or supplemented fatty acids and cell formation. This is a still underestimated point in the experimental design of PUFA supplementations which are used for both in vitro and in vivo experiments, to study their benefits for health and diseases. Moreover, the possibility of an inadequate PUFA intake from foods is realistic, also because knowledge of the importance of lipids is not diffuse. The so-called essential fatty acid (EFA) deficiency is nowadays a condition that is highlighted at a global level [13]. EFA deficiency is also reported in the MSD manual [14], but PUFA levels in the body are not a required check-up in clinical practice. On the other hand, without appropriate and balanced EFA intakes, organisms cannot work, with the consequences of a deficit having symptoms that are quite similar to a pathology. As a simple example, skin problems such as dermatitis are one of the most diffuse symptoms of an EFA imbalance [15]. In clinical studies, where fatty acid supplementations are used, the fatty acid profiles of the patients are not determined at the start, so it is not known whether the dosage or the type of fatty acids used are appropriate, either to fix the deficiency or, instead, to realize an excess or unnecessary supplementation. It is worth noting that a precise fatty acid distribution occurs in tissues [16,17], therefore an excess of fatty acid supplementation brings the possibility of excessive incorporation in the lipid pools, with creation of an imbalance. Table 1 shows the distribution of fatty acids of the four families (SFA, MUFA, omega-6 and omega-3 PUFA) in some human tissues. Clearly, high dietary intakes of SFA or insufficient intakes of PUFA can be reflected in tissue imbalance, altering the correct compositions with respect to the other fatty acid families, and this may result in tissue disfunctions [16,17].

Only recently, the consequences of an excess of the omega-6 linoleic acid (9c,12c-18:2), which is the most prevalent PUFA in Western diets compared to the other EFAs, were reported and was questioned as to its possible toxicity at brain tissue level [18]. It is interesting to see, in Table 1, the low percentage of linoleic acid in the brain.

An excess of linoleic acid can favor the increase in the omega-6 metabolic cascade shown in Figure 1, which has a crucial importance for the lipid signaling related to cancer, especially concerning inflammatory and proliferative processes, that depart from arachidonic acid and its eicosanoid metabolites [19]. The loss of control of inflammatory signals can also be due to a scarce intake of omega-3-related foods, therefore an important primary prevention strategy is nowadays clearly connected with the balance between omega-6 and omega-3 food intakes. For the biological effects of omega-3, it is important to note that bioactive lipids such as resolvins, protectins and marensins can be generated from EPA and DHA. Intense research is ongoing in order to unveil the molecular mechanisms of immune and anti-inflammatory activities which can be used as a support in cancer therapies [20].

In cancer, a clear disruption of the lipid balance in cell compartments was evidenced, in particular in cell membranes which are deeply involved in the propagation of growth and proliferation signaling. Beside the role of membrane-bound cholesterol in the onset of several diseases, including cancer [21], cancer cells adapt themselves rapidly by re-organization of their plasma membranes to preserve proliferation, escape apoptosis and resist anticancer drugs treatment [22,23]. The molecular mechanisms played by fatty acids in cancer were treated in several reviews, where readers can deepen their knowledge of these subjects [24,25,26]. Lipids are not a sole matter of energy for cancer cells, but mainly the regulators of the signaling asset, needed for sustaining cancer cell metabolism. In cancer the fine tuning of the membrane properties by fatty acids can exert a bi-directional effect: (i) due to an increased delta-9 desaturase activity and formation of MUFA, membrane fluidity results in being increased and this is a *conditio sine qua non* that signaling cascades work (for example, protein kinase B/mammalian target of rapamycin (Akt/mTOR)) and activate growth [27,28,29,30]; (ii) due to an increased membrane thickness, connected with the re-organization of lipid rafts and proteins, proteins change their activity, as in the case of osmo-sensors and the oncogenic Ras (Rat sarcoma virus) cascade [31] or their presence can even change, with dramatic consequences for the overall membrane organization and dimensions [32,33].

## 3. The Role of Lipogenesis and Ketogenic Diets in Cancer

The fatty acid contributions of food (or supplements) for cancer therapies can play important roles in the lipogenesis process. The SFA–MUFA pathway is endogenous, starting with the enzymatic complex of fatty acid synthase (FAS), which is more activated in cancer, and the production of palmitic acid. Consequently, its transformation to stearic acid and then to oleic acid, the most important MUFA in eukaryotic cells, is a faster process in cancer cells (see Figure 1) [24,27]. It is worth noting that other enzymes also show excessive activation, thus accelerating the lipogenesis process. Examples are citrate lyase, which is overexpressed to form acetyl-CoA from citrate formed in the Krebs cycle [34]; similarly, acetyl-CoA carboxylase, is positively regulated by PI3K (Phosphatidylinositol 3-kinase)/AKT/mTOR signaling, favoring the formation of malonyl-CoA from acetyl-CoA and resulting in a key to initiate fatty acid synthesis [35]. In addition, glucose assumption and metabolism are connected for the formation of pyruvate, which is the starting point of de novo synthesis of palmitic acid. The role of lipogenesis in cancer suggested that modification of patients’ diet can synergize with cancer therapy. Recent reviews and scientific debates are available in literature on this subject [36,37]. At this point it is worth underlining the role of ketogenic diets (KD). KD provide less than 20% of the caloric intake from protein and carbohydrates, and the rest of the energy is provided by fats. What are the approaches used in KD? KD implies a dietary restriction of glucose which induces a fasting mimicking state and a nutritional ketosis. KD was used in various human diseases, the most noteworthy being its successful use for the treatment of intractable epilepsy [38]. In cancer, there are relevant positive evidences from preclinical studies, showing that KD can reduce tumor growth and increase survival time [39]. The molecular mechanisms responsible for these effects are possibly related to the creation of an unfavorable tumor microenvironment (reviewed elsewhere [40,41,42]). However, evidence in human trials is more unclear, since few and small clinical trials were carried out, with little concordance among results that can be observed [39,40]. Recent systematic reviews agree in finding no conclusive evidence of KD effectivity for anti-tumor effect or increased survival [43,44]. Furthermore, a recent metanalysis of randomized clinical trials reviewed studied parameters and compared KD vs. a control diet. Most of the measured biochemical parameters (total, HDL and LDL cholesterol, triglycerides, body weight, fasting blood glucose and insulin) were not statistically significant when comparing the control diet group and KD group, except for ketosis, that was higher in the KD group (risk ratio (RR) = 3.58, 95% CI = 1.36, 9.40), and level of satisfaction, that presented better values in non-KD (results from only one study, Std mean difference (95% CI) = 1.52 (0.47, 2.57) [42]. Not all of the studies analyzed the same parameters, there were differences in the definition of the diets, the intervention length was variable, and the reviewed trials were in different cancer types, which could explain the lack of conclusive results. Interestingly, in all of the included studies the completion rate of the participants was always lower in the KD diet (completion rate 45 to 75% of participants enrolled) compared with the control diet, reflecting some difficulties in sticking to the KD [42]. The described lack of conclusive results could be explained by the heterogeneity of the studies carried out so far, since there were differences in the definition of the diets, the intervention length and the cancer types studied, among other disparities and limitations.

It is relevant for the subject of this review that none of the KD studies proposed so far present details of ratios or amounts of SFA, MUFA or PUFA in the diets. Based on the previously discussed roles of fatty acids in Section 2, such imprecise fat strategy cannot be understood.

Due to the activation of lipogenesis, an increased formation of palmitic and stearic acids could cause membrane rigidity and disfunctions, conditions that should bring cells to death. However, high levels of SFA induce activation of delta-9 desaturase enzyme (stearoyl CoA-desaturase), thus moving the target of a fatty acid-based strategy to stopping the transformation of stearic acid to oleic acid. As a matter of fact, the pharmacological inhibition approach resulted in being difficult to apply, since the absence of adequate MUFA content in the membranes produces toxicity not only to cancer cells, but also to normal ones [29,30,45,46]. A recent study pointed out that metastasis-initiating cells particularly rely on dietary lipids to promote metastasis. These metastasis-initiating cells are characterized by a high cell-surface expression of the FA receptor CD-36 and are shown to be responsible for initiating metastasis in orthotopic models of human oral squamous cell carcinoma, and also in experimental metastasis models of human melanoma and breast cancer. A high fat diet and stimulation with palmitic acid were able to boost metastatic potential in these metastatic models [47]. In 2021, the same research group discovered that dietary palmitic acid, but not the MUFA oleic acid nor the omega-6 linoleic acid, promotes metastasis in oral carcinomas and melanoma in mice [48]. More recently, studying mechanisms underlying obesity promotion for stem cell-like properties of breast cancer cells, cellular adaptation to obesity was found to be governed by palmitic acid, leading to enhanced tumor formation capacity of breast cancer cells [49].

At this point it is worth raising interest and awareness of researchers in the field of dietary fatty acids and their effects in cancer models, that the use of palmitic acid in the cell models also requires deepening understanding of two aspects: (i) the effects due to the contemporaneous presence of other fatty acid types, as it happens in diets, (ii) the time of exposure of the cell cultures to the dietary conditions. As an example, it was shown that the neuroblastoma cell line (NB-100) treated with 150 µM palmitic acid underwent caspase-mediated apoptosis, whereas when using 50 µM oleic, 50 µM arachidonic and 150 µM palmitic acids an interesting change of the cell fate occurred, with the membrane lipidome remodeling combining with suppression of the caspase activation and maintenance of cell viability [50].

On the other hand, dietary interventions were not yet directed to regulate the intake of specific fatty acids. Among registered clinical trials with cancer patients, not a single nutritional intervention can be found with the objective of regulating the intake of singular SFA or MUFA, while several interventions were registered testing healthy, Mediterranean or ketogenic diets effects on cancer patient prognosis, survival and/or wellbeing (ISRCTN registry, EU Clinical Trials Register, Clinicaltrials.gov). This gap can be explained by the fact that fatty acids, such as palmitic or stearic acids, with (positive and negative) impacts on cancer are commonly present in the same food sources, which makes it unviable to control the intake of one without affecting or altering the intake of the other [51]. Still, it would be possible to reduce somehow the dietary intake of palmitic acid avoiding the main source of this fatty acid, such as palm oil (and derived products). A Table with the levels of total fats, as well as the palmitic and stearic acids, in foods is presented (Table 2) using the values found, when possible, in two different databases of foods issued in USA and England. Obviously, the reported fatty acid contents should be considered an average of those contained in foods from other territories, but still readers can have an idea of the SFA contents and increase their awareness.

Taken together, this knowledge opens the opportunity to therapeutic approaches for cancer patients that include targeted nutrition, considering specific dietary recommendations for food products based on their fatty acid composition.

In the frame of the Mediterranean diet and the use of extra virgin olive oil (EVOO), it was also proved that the use of oleic acid could be beneficial for the presence of small quantities of anticancer molecules (oleocanthal, terpenes, squalene and antioxidant molecules) [52], and also to the presence of oleic acid itself. EVOO regular intake in the diet was considered overall to influence the reduction of cancer incidence and also favor an improved survival [53].

Regarding dietary patterns interventions in cancer patients and survivors, a recent review and metanalysis highlighted that there is a good evidence of the effect of healthy diets on overall survival and quality of live for survivors of certain cancers, such as breast and colon cancer [54]. Although variability exists among studies and healthy diets’ definitions, some aspects are common, such as increased intakes of fruit, vegetables, and fish and reduced intake of saturated fats. It is worth recalling that the dietary insufficiency of these foods, or other PUFA-containing foods, such as nuts and seed oils, together with an increase in SFA-containing foods (see Table 2), are claimed to be among the 15 reasons of risk or death for non-communicable diseases in the Global Burden of Disease Study, which gives a comprehensive overview of 27 years follow-up in 195 countries [55]. From this, it should derive that not only should diet be more and more used in prevention, but certainly a single cancer patient should not remain without molecular profiling and dietary directions.

An interesting advancement was recently published on the SFA-MUFA transformation by the desaturase process, reconsidering the whole scenario of the double bond insertion in a SFA molecule. It was found that the double bond can be inserted in SFA, not only in the delta-9 position (such as in palmitic and stearic acids, with the synthesis of palmitoleic and oleic acids, respectively, members of the omega-9 family), but it can also involve delta-6 desaturase enzyme [56,57,58]. It is well-known that the main activity of delta-6 desaturase occurs for the transformation of essential PUFAs (with the first step of the transformations of linoleic and alpha-linolenic acids); however, it was recently found that delta-6 desaturase can work on palmitic acid producing the omega-10 sapienic acid (6c-16:1). This novel pathway opens another interesting hypothesis for the fatty acid-based mechanisms due to dietary lipids: as shown in Figure 2, when palmitic acid has increased intake/formation, as occurs in cancer, a higher formation of sapienic acid can take place, and this anomalous metabolism can be further helped by a decreased PUFA presence or intake. Interestingly, proof was already provided that sapienic acid is transformed by delta-5 desaturase into sebaleic acid (5c,8c-18:2), which is so far the only PUFA endogenously produced by our cells [57,58].

Lipidomic analysis are very important for the follow-up of SFA-MUFA pathways. It is worth noting that the use of mass spectrometry tools for analyzing fatty acids can fail in its recognition of sapienic acid, which has the same molecular mass of palmitoleic acid (*m*/*z* = 254), whereas GC is the gold standard to recognize and quantify properly these two metabolites (in GC/MS as methyl esters with *m*/*z* = 268) [1,56]. The novel connection between essential PUFA and lipogenesis to produce omega-10 (or n-10) fatty acids and, consequently, disrupt the usual metabolism will be explored using different diets, to provide more insights on the significance of the fatty acid balance, and to suggest new synergic fatty acid-based strategies combining foods and nutraceuticals in cancer therapy.

Knowledge of the fatty acid metabolism is needed to properly address cancer therapy. The pharmacological intervention with inhibitors of lipid enzymes can have an insufficient control on the cancer growth [59]. In these cases, it would help to keep in mind the alternative delta-6 desaturase pathway giving relevance to the follow-up of sapienic acid metabolism. More recently, in cancer cells, it was shown that sapienic acid supplementation induces changes in growth factors and proliferative signaling [60].

## 4. Fatty Acid-Based Membrane Balance for Signaling

From the biological and mechanistic points of view, fatty acids are activated after detachment from membrane phospholipids by phospholipase enzymes, in particular of phospholipase A_2_ (PLA_2_), releasing the fatty acid in position *sn*-2 of L-glycerol (Figure 3).

The liberated fatty acid is then present in the cytoplasm and transformed intracellularly by enzymatic processes to give a plethora of bioactive lipids. It can be said that levels of omega-6 and omega-3 PUFA in membranes generate levels of eicosanoid and docosanoid metabolites upon stimulation [8]. Such bioactive lipids are potent biological mediators at fento- or nano-molar concentrations and responsible for the inflammatory and anti-inflammatory response in a variety of health conditions. As an example, in asthma patients the levels of red blood cell (RBC) membrane omega-6 fatty acids were found to positively correlate with the higher plasma levels of eicosanoids (PGE_2_, TXA_2_, LTB_4_, PGE_1_, 6-k-PGF_1**α**_ and PGF_2**α**_) compared to healthy controls [61]. This result can suggest dietary intervention to balance the omega-6 intakes and lower the inflammatory response. Indeed, nutrition or supplementation of fatty acids can influence the quality of the fatty acid residues in membrane phospholipids, thus, exerting a control of the free fatty acids liberated by the PLA_2_ process and, consequently, of the generated bioactive lipids.

Figure 3 represents the different fates of the omega-6 di-homo-ɣ-linolenic and arachidonic acids (DGLA and ARA), and of the omega-3 eicosapentaenoic and docosahexaenoic acids (EPA and DHA) in the generation of lipid mediators. Cancer cells’ proliferation and invasiveness are favored by an altered balance among PUFA, as evidenced by fatty acid biomarkers, that can be obtained from the analysis of the red blood cell membrane lipidome [1,24].

Targets of the bioactive lipids, such as PGs, involves receptors and pathways for cancer signaling and their levels are known to increase in tumors. For example, PGE_2_ levels increase in the case of osteosarcoma, the most frequent primary bone tumor. It was found that tissues drawn from patients showed a high expression of the staphylococcal nuclease domain containing 1 (SND1), which is a protein expressed in multiple cancers inducing angiogenesis, and induces PGE_2_ increase through the nuclear factor NF-κB/cyclooxygenase (COX)2 pathway [62]. In osteosarcoma PGE_2_ is also able to interact with E-series prostaglandin (EP)1 receptor inducing cell proliferation and decreasing cell apoptosis [63]. On the other hand, balanced signaling is known to occur by prostaglandins derived from different eicosanoid precursors. DGLA produces PGE_1_ by interaction with COX enzymes (see Figure 3) resulting in anti-inflammatory activity. This prostaglandin is also known to have antitumoral activity for its ability to interact with EP4 receptor, as shown for myeloproliferative disorders, such as chronic myelogenous leukemia (CML) [64]. The complexity of this subject was treated by a recent review [65], where readers can find proper and complete information.

Based on the above scenario, the possibility of targeting fatty acid metabolism in cancer by appropriate fatty acid-based food intakes is realistic and use the membrane lipid therapy to improve the efficacy of pharmacological therapies, contrasting pro-inflammatory and pro-angiogenic lipids by analogous transformations that produce anti-inflammatory and anti-proliferative lipids [11,66].

What is the basic mechanism by which fatty acid supplementation can be effective for changing the membrane composition? The answer comes from an important and natural process that intervenes for the maintenance and turnover of the membrane lipid assembly. This process is called “lipid remodeling” and was studied in the 1950s by Lands [67,68]. As shown in Figure 4, the membrane phospholipid composition is under continuous change, due to the activity of two classes of enzymes: phospholipases (PL) and lysophospholipid acyl transferases (LPAT), and such remodeling in cancer can be used as a membrane strategy for disease control [69,70].

## 5. Biodistribution and Composition of Fatty Acid-Based Supplements

Fatty acid supplementation participates with natural and spontaneous processes occurring in the body, such as membrane formation, lipid remodeling and membrane-based signaling, as previously discussed. After oral supplementation, fatty acid bioavailability and turnover can be followed-up, similarly to pharmacological treatments. The first step is to establish where in the body the follow-up can be performed in order to express fatty acid bioavailability. Measures of fatty acid levels are often performed in blood plasma lipids. This is informative of the adsorption process that brings lipids in circulation from the intestine, but such levels can hardly, and only indirectly, be correlated to the biological effects which do not occur in plasma. Indeed, to estimate the bioavailability of fatty acid supplementation it is more informative to measure the levels of fatty acids in tissues, or at least after metabolic processing: the fate of metabolized fatty acids are either to be incorporated as phospholipids at the level of one of their relevant “active sites”, which are membranes, as described above, or to generate bioactive lipids, whose levels can be measured in plasma. Fatty acid supplementation, which usually means to supply triglycerides contained in natural vegetable or animal sources, undergoes processing by the lipase enzymes and the resulting fatty acids form part of the lipid pools, mainly as acyl-CoA derivatives, as shown in Figure 4. In the form of acyl-CoA, fatty acids can enter the remodeling pathway for membrane incorporation, and from this latter location can generate bioactive lipids, as shown in Figure 3 and Figure 4.

It is worth noting that, in the case of high dosage supplementations, triglycerides tend to accumulate in lipid droplets, also present in the intra-cellular environment. After consideration of the balance in lipid pools, it also becomes clear that the use of natural vegetable or animal oils (sources of triglycerides and, then, of fatty acids) in the formulations must be adequate in quantity and quality to fulfill such balance. Consequently, fatty acid-based foods and nutraceuticals must be evaluated for their whole fatty acid composition, including SFA, MUFA and PUFA. In the nutrition facts’ labels of marketed products, only total lipids and the relative percentage of SFA are reported so far. The additional information of the specific fatty acid types, such as in the case of omega-3 supplements, report only the quantities of EPA and DHA per gram of oil. This is an unclear information for consumers about the quality and quantity of the fatty acids in the formulation. Examining the fatty acid composition of marketed products, advertised as “source of omega-3”, it was determined that in several cases the “claimed” omega-3 PUFAs were not even the prevalent fatty acid type, but constituted only 20–30% of the total fatty acid composition (Table 3) [71]. The composition is important information, since in the biodistribution of an omega-3 supplement containing 30% of SFA, 50% MUFA and 11% of PUFA (of which 9% omega-3), it is highly unlikely to expect that the fatty acids to be prevalently distributed are the PUFA ones. Remarkably, the content of unnatural fatty acids as trans fatty acids (TFA), deriving from the fish oil deodorization, is completely ignored [71].

It is also evidenced that, in human and animal studies, the most commonly studied fish oil is anchovy oil, which typically contains about 30% of EPA and DHA, in a 18/12 ratio (180 mg of EPA and 120 mg of DHA per g of oil), and 70% of the supplemented fats are SFA and MUFA. The fatty acid-based formulations are an important subject of research, and more work is needed in the field of PUFA-containing natural sources, also for the biotechnological development of specifically omega-enriched vegetables and animal products [72,73,74,75]. As a matter of fact, when seeking new ingredients and formulations, it is necessary to follow the principle that oils must contain prevalently the specific fatty acid molecule, which means that not less than 60% of the “active” ingredient should be present in the total weight of the active molecules that compose the food or nutraceutical formulation.

Finally, as explained above, when a fatty acid-based food or nutraceutical is used, the levels of incorporation in membrane phospholipids is a safe information to be acquired in order to evaluate both the effectiveness of the supplementation and the correct insertion in the fatty acid turnover. In this respect, the measurement of the incorporation of the supplemented fatty acid in the RBC membrane lipidome, as both “circulating tissue” and representative cell for all of the tissues [1,24], can be considered as an established methodology to insert in the validation of a supplementation.

## 6. Applications of Fatty Acids as Nutraceuticals to Cancer Therapy: Assignment vs. Personalization

The specific activities of fatty acids in cancer were discussed in the previous sections and also in a recent review by our groups [24]. The use of a fatty acid-based strategy to adjuvate the pharmacological therapy is being considered more and more, due to evidence-based medical reports. Omega-3 fatty acids were the first to be focused on in cellular and animal models, and the good results obtained in anticancer activity motivated the translational use to humans [76]. By various mechanisms, that include peroxidation reactivity, omega-3 PUFA triggers cancer cell apoptosis, and synergizes to increase the sensitivity of tumor cells to conventional therapies, with interesting applications to cancers resistant to treatment [77]. Interestingly, in contrast to traditional therapies, omega-3 appears to cause selective cytotoxicity towards cancer cells with little or no toxicity on normal cells. It was also observed that omega-3 supplementation can improve the condition of tumor patients with weight loss [78,79]. The results of omega-3 trials were well described in reviews, where readers can have a broader presentation of the subject [80,81].

Among the omega-3 elements, docosahexaenoic acid (DHA) showed the best results in reducing tumor growth in preclinical models, also when combined with chemotherapy, and is known to beneficially modulate the systemic immune function. The effects of DHA to mediate the induction of apoptosis/reduction of proliferation in vitro and in vivo, as well as on cell cycle regulation in cancer cells, are also demonstrated [82]. It must be evidenced that the use of a high-DHA containing source allowed for a clear differentiation of the DHA biological effects from the EPA+DHA sources [83]. The main molecular mechanisms involved in such activity were also addressed for DHA [84,85,86], and also in several DHA-anticancer couples, such as with sorafenib and paclitaxel [87,88], as well as by comparing effects of antitumoral with and without DHA [89]. It is necessary to recall at this point the previously discussed omega-3 contents in supplements, therefore in cancer the high DHA content becomes the most important requirement to ask of the formulation, coupled with the proof of the biodistribution to the membrane active site.

At this point, it is worth raising attention and awareness on focusing research on cancer only related to a specific type of fatty acid, such as shown for the omega-3 fatty acids. Here, we wish to remark again that omega-3 fatty acids are one of the four families constituting the cellular pool of fatty acids, and in cancer it is necessary to focus on the signaling balance, as explained in Section 3. In particular, concerning the choice of PUFA supplementation, it is worth noting that omega-3 fatty acids are not the only ones to work for tumor signaling depression: one omega-6 fatty acid—namely, di-homo-ɣ-linolenic acid (DGLA)—has high potential against proliferation diseases [90], based on the competitive activity in cellular lipid metabolism and eicosanoid biosynthesis by cyclooxygenase and lipoxygenase enzymes. For example, DGLA can be further converted by these enzymes to 15-(S)-hydroxy-8,11,13-eicosatrienoic acid (HeTE) and prostaglandin E_1_ (PGE_1_). PGE_1_ counterbalances PGE_2_, obtained by the arachidonic acid transformation, with PGE_2_ being able to mediate inflammation whereas PGE_1_ acts as an anti-inflammatory factor (see Figure 3). Nowadays, the omega-6 cascade cannot be considered to have exclusively an inflammatory effect because of arachidonic acid, but the roles of the omega-6 precursor linoleic acid, and of the omega-6 metabolites, ɣ-linolenic acid (GLA) and DGLA, were defined specifically as anti-inflammatory [91]. Moreover, deepening the roles of prostaglandins and EP (E-series prostaglandin) receptors, a family of G-protein coupled receptors designated as EP1–3 and EP4, the differentiation of omega-6 effects in cancer signaling became clearer and clearer [65], highlighting the use of GLA sources as supplementation in cancer treatments. GLA-enriched sources are the oils from the seeds of well-known plants, such as evening primrose and borage, as well as from the parts (especially stems) and seeds of less known plants, which are found wild, such as purslane [92].

From the overview of research in fatty acid and cancer, the “precision nutraceuticals” strategy becomes evident and starts with the personalization of the fatty acid-based food and supplementation assignment, depending on the specific condition of the fatty acid pool in the body. The fatty acid-based membrane “fingerprint” in tumoral patients can be used to assign “tailor-made” fatty acid-based dietary and supplement treatments and to follow-up metabolism over time in a non-invasive way. Fatty acid-based membrane lipidome analysis has reached a good technology readiness level, due to the availability of high-throughput equipment and an accredited (quality standard according to the ISO/IEC 17025) protocol for the isolation of mature RBC membrane phospholipids, used in various studies of patients in various conditions, including cancer [1,93,94,95], overcoming tedious and imprecise manual operations. The mature RBC membrane profile contains well-recognized fatty acid biomarkers for cancer [24] and can help to individuate the patient’s need in view of creating the best lipid environment to sustain normal cells and defeat cancer ones. The membrane lipidome profile can help the patient’s follow-up during associated fatty acid-chemotherapy approach, controlling the levels of PUFA and the effects on lipids that are precursors of the proliferation and lipogenesis signals, as previously discussed. We can also envisage that the membrane profile and remodeling by membrane lipid therapy becomes strategically important in combination with early diagnostic tools, which are proposed for prognostic screening and prevention of cancer, such as circulating cancer cells [96,97]. Literature evidence in favor of the holistic vision in cancer suggests such a development to be strategic for early cancer diagnostic approaches, at a stage where chemotherapeutic treatments cannot be prescribed to the patient, whereas diets can be important tools for active prevention, including fatty acid-based strategies [98].

## 7. Fatty Acid Peroxidation as Molecular Reactivity in Cancer Therapy

Reactive Oxygen Species (ROS), originated from endogenous source, such as mitochondrial respiration, as well as exogenous sources, such as xenobiotics and radiations, are formed in various diseases and also in cancer [99,100]. Fatty acids are involved in chemical and enzymatic peroxidation processes [101,102] which have an impact on lipid pools, as well as on membrane phospholipid assets, with changes in the structural and functional properties [24,103,104]. During chemical peroxidation, almost 150 species can be formed [105]; lipid hydroperoxides, the primary compounds, are unstable and tend to decompose to more stable but still reactive and potentially toxic secondary species, mainly the aldehydes, capable of diffusing and creating adducts with amino-containing DNA bases, or cross-links using thiol- and amino-functionalities in proteins; as a matter of fact, malondialdehyde (MDA), 4-hydroxy-hexenal (4-HHE), 4-hydroxy-2-nonenal (4-HNE), and acrolein (ACR) can produce adducts to DNA and proteins that are detected in several cancers [106,107,108,109]. These damages can impair the functioning of normal cells and are evoked as important factors in tumorigenesis. ROS can stimulate tumorigenesis by activation of signaling cascades, modulating growth factors, angiogenesis, migration, metastatic processes and chemo-resistance [110,111,112,113,114]. For example, MDA originating from the peroxidation of arachidonic acid or DHA remains one of the main mutagenic lipid peroxidation products, forming adducts with the guanine moiety of DNA, whereas 4-HNE is the final product of the cardiolipins’ oxidation in mitochondria [115,116]. Acrolein is the strongest electrophile among the previously mentioned aldehydes interacting with cellular nucleophiles, enzymes and proteins. It can interact with DNA and induces cell death for its cytotoxic effect [117].

In this scenario, it is important to evidence that fatty acid peroxidation rate is low in physiological conditions, and this can stimulate cell survival by activation of the antioxidant response. On the other hand, a high peroxidation rate, and consequently more extended damages, induce the death accelerating apoptosis, and the necrosis processes [118,119]. The levels of end-products of fatty acid peroxidation coming from the chemical pathway can be used as a biomarker of oxidative stress, as shown in some epidemiological cancer studies summarized in Table 4 [120,121,122,123,124,125,126,127,128,129,130,131,132]; for each lipid biomarker, the main products, their detection in biological specimens and the dose-dependent molecular effects are indicated.

PUFA are also enzymatically peroxidized by peroxidases or by proteins having a peroxidase activity, such as lipoxygenases (LOX), cyclooxygenases (COX), and cytochrome P450, or by enzymes or hemoproteins showing a peroxidase activity only in specific conditions, such as cytochrome C binding to cardiolipins [133,134]. Such products have different activities if they derive from omega-3 or omega-6 fatty acids, as discussed for prostaglandins in Section 6, and reported in several reviews [135,136,137].

In this review we wish to underline that fatty acid-based foods and supplements, especially those containing PUFA, can be helpful for their peroxidation reactivity in cancer therapies with drugs as well as in radiotherapy, trying to overcome the antioxidant defense of cancer cells and induce cell death [138]. It is worth recalling that synthetic drugs such as cisplatin, bleomycin and doxorubicin, have a ROS-dependent anticancer effect that can be coupled with an enhanced pro-oxidant environment created by highly unsaturated molecules such as PUFA, but also by metal complexes and antioxidants, and even by an increase in physical activity [139,140,141,142,143]. In such fatty acid-based strategies, precision tools must be used to ensure success and control along the treatment, monitoring both the patient’s fatty acid profile, as explained in Section 6, and the peroxidation products, as shown in Table 4. The events of peroxidation and apoptosis have evolved recently into the ferroptosis process which is an alternative means of cell death, different from apoptosis and necrosis [144], determined by an iron-dependent high accumulation of lipid hydroperoxides; cell death is triggered by unrestricted lipid peroxidation and, consequently, cell membrane disruption. Fatty acid peroxidation in the ferroptosis process is mainly due to the inactivation of glutathione peroxidase 4 (GPX4) [145] that catalyzes the conversion of toxic lipid hydroperoxides in non-toxic lipid alcohols. Therefore, the ferroptosis process can be altered by iron chelators, lipophilic antioxidants and also the depletion of polyunsaturated fatty acids, due to poor repair and turnover [146]. Here it is not the scope to review the ferroptosis process that can be deepened by literature survey [147,148,149]. The beneficial or detrimental effects of antioxidants during cancer therapy, based on the free radical mechanism, is a debated topic [150,151,152]. Problems can arise from the fact that antioxidant supplements, recommended to tolerate or mitigate adverse side effects of oxidative damages induced by treatments, can reduce or abolish the efficacy of such therapies. Indeed, lipid peroxidation induced by anti-tumoral treatment generates oxidation by-products from unsaturated fatty acids, that can act as adjuvants and supporters of cytotoxicity in cancer therapy, but these by-products are annihilated in the presence of antioxidants and peroxidation inhibitors, such as vitamin E, N-acetyl cysteine and *α*-tocopherol [121,153,154].

This subject is still the matter of studies and debates to reach general recommendations, because final antioxidant effects depend on several variables, such as the type of tumor, the mechanism by which therapies function, the type and dosage of supplement, as well as the patient’s antioxidant status [155]. Types and doses of antioxidant supplements must be individuated clearly, in order to be able to promote the protective effects on normal cells without invalidating the drug-induced free radical toxicity on cancer cells. Actually, prescription of antioxidants, such as vitamins and supplements, is very moderate during radio- and chemotherapies [156,157].

## 8. Conclusions

In summary, the success of fatty acid-based food and, especially, supplementation strategies in cancer therapy is connected with a delicate balance which involves: (a) the quality and quantity of fatty acids in the cellular pool and, in particular, in membranes; (b) the oxidative and anti-oxidant processes in order to sustain the adjuvant effect of free radical and ROS mechanism. As discussed in this review, the scenario of synergic effects induced by fatty acids for the cancer cell growth control is complex and a fine monitoring of the individual status is needed, in order to assign the specific treatment and fully control the benefits of lipid therapy.

In conclusion, fatty acids emerge as crucial molecules involved in structural and functional roles at cellular level, as well as in nutritional directions and supplementations, with a high potential to be coupled with cancer disease evaluation and intervention. As shown in this review, the importance of lipids and fatty acid types is established, distinguishing saturated from mono- and poly-unsaturated fats, also from a global perspective [55], however, in nutritional epidemiology it is still highly debated as to how foods increase cancer risk [158] or support treatments and counteract malnutrition [37]. Cell membrane lipidome emerges as a target for cancer studies, in order to evaluate fatty acid quantity, quality and bioavailability, in particular in forming membrane phospholipids and functioning for a physiological balance. Indeed, we believe that the membrane fatty acid-based approach of precision nutrition and nutraceuticals in cancer and prevention is highly motivated by both the needs of cells, especially of tumoral cells, for membrane lipids to ensure replication, and the creation of a controlled and anti-inflammatory molecular environment, which can then cooperate with other preventive measures, as well as with therapies.

## Figures and Tables

**Figure 1 ijms-23-06030-f001:**
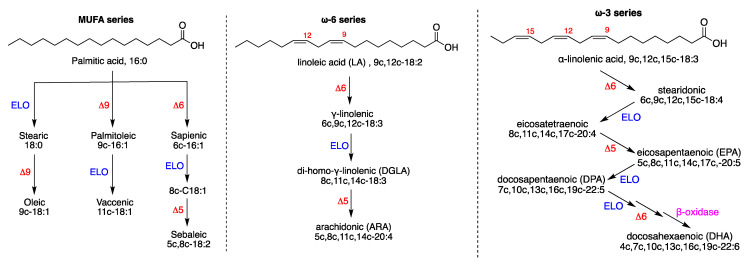
Formation of unsaturated fatty acids: (**left**) biosynthetic pathways of MUFA starting from palmitic acid (SFA); (**center**) the omega-6 PUFA biosynthesis starting from linoleic acid; (**right**) the omega-3 PUFA biosynthesis starting from alpha-linolenic acid. Enzymes: ELO elongase; Δ5-, Δ6-, and Δ9-desaturase; β-oxidase. Numerical abbreviations describing the position and geometry of the double bonds (e.g., 9c), the notation of the carbon chain length and total number of double bonds (e.g., C18:2); in parenthesis, the used acronyms (e.g., ARA for arachidonic acid). Reprinted from Ref. [10]. Copyright year 2022, Ferreri et al.

**Figure 2 ijms-23-06030-f002:**
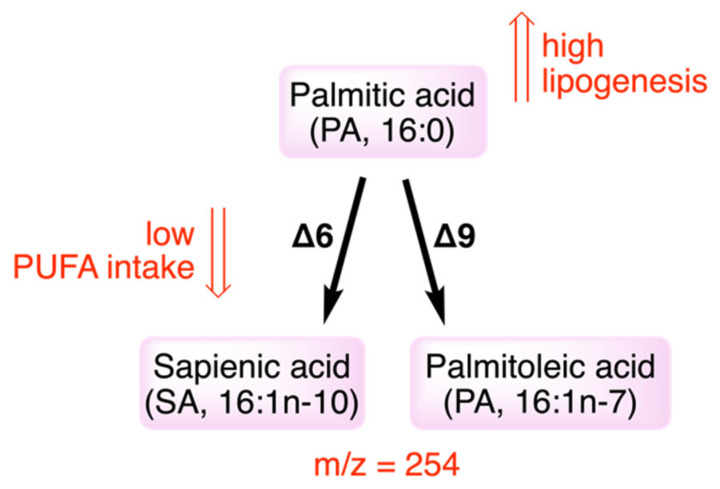
Palmitic acid transformations by delta-9 and delta-6 desaturase enzymes. Positional isomers formation from delta-9 and delta-6 desaturase activities on palmitic acid: sapienic acid has the same molecular mass of palmitoleic acid (*m*/*z* = 254 or their methyl esters *m*/*z* = 268). For low PUFA intakes, delta-6 desaturase can enter in the transformation of palmitic acid to sapienic acid which is a n-10 fatty acid not belonging to the usual metabolism of saturated fatty acids.

**Figure 3 ijms-23-06030-f003:**
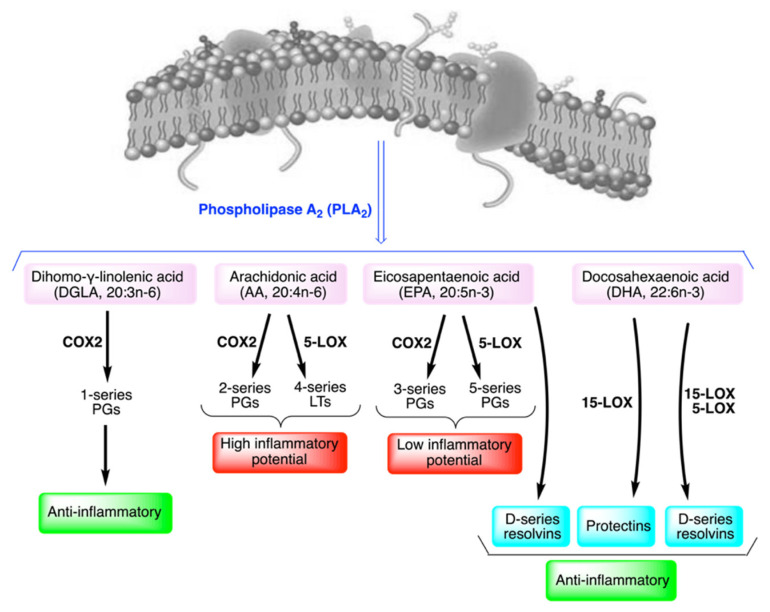
The balance among omega-6 and omega-3 fatty acids in membranes for the generation of signaling molecules upon release from membrane phospholipids.

**Figure 4 ijms-23-06030-f004:**
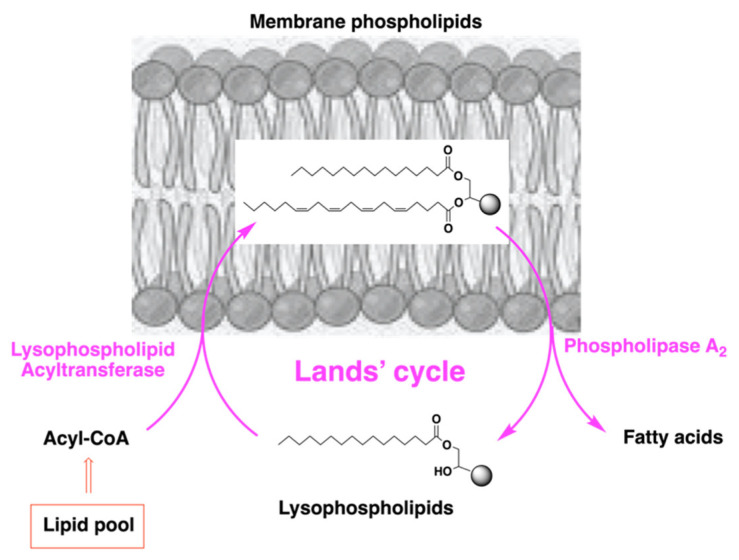
The remodeling mechanism for membrane phospholipid turnover.

**Table 1 ijms-23-06030-t001:** Details of the fatty acid composition (% rel.) of membrane phospholipids in human tissues ^1^.

Fatty Acids	Adipose Tissue (% rel.)	Red Blood Cell (% rel.)	Liver (% rel.)	Retina (% rel.)	Brain (% rel.)
9c,12c-18:2 ω-6	10.5	9.3	17.5	1.4	0.6
5c,8c,11c,14c-20:4 ω-6 (ARA)	0.3	15.2	7.7	9.6	7.7
20:3 ω-6 (DGLA)	0.2	1.5	1.6	n.d.	1.2
20:5 ω-3 (EPA)	traces	0.7	0.4	0.1	traces
20:6 ω-3 (DHA)	0.3	3.2	3.4	19.7	7.2
SFA	27.2	43.1	42.0	48.2	45.9
MUFA	59.7	23.0	23.8	14.2	29.7
PUFA	13.1	33.3	32.0	37.2	23.4
ω-6/ω-3	0.17	0.21	0.17	1.32	0.46

^1^ Data Taken from ref. [16].

**Table 2 ijms-23-06030-t002:** Content of total fats, palmitic acid and stearic acid in foods expressed as g/100 g of food (edible portion). Source of data: ^1^ U.S. Department of Agriculture. Food Data Central Data obtained from The National Nutrient Database for Standard Reference Legacy Release, and ^2^ UK—McCance Widdowson’s Composition of Foods 2021.

Food	Total Fat ^1,§^	Palmitic Acid ^1,§^	Stearic Acid ^1,§^	Total Fat ^2,§^	Palmitic Acid ^2,§^	Stearic Acid ^2,§^
OILS AND FATS						
Palm oil	100.00	43.5	4.3	99.90	41.8	4.6
Linseed oil	100.00	5.11	3.37			
Sunflower oil	100.00	5.9	4.5	99.90	6.2	4.3
Olive oil	100.00	11.3	1.95	99.90	10.1	3
Soy oil	100.00	11.2	12.6	99.90	10.7	3.8
Coconut oil	99.10	8.64	2.52	99.90	8.4	2.5
Corn oil	100.00	10.6	1.85	99.90	11.3	2.1
Canola oil	100.00	4.3	2.09	99.90	4.2	1.5
Margarine (regular, from various fats)	80.20	7.08	5.81	76.40	20.8	2.3
Butter	81.10	21.7	10	82.20	22.4	8.6
Lard	100.00	23.8	13.5	99.00	24.4	14.1
NUTS						
Peanuts	49.70	3.98	1.2	49.80	4.71	1.19
Hazelnuts	60.80	3.1	1.26	63.50	3.16	1.09
Walnuts	65.20	4.4	1.66	68.50	4.91	1.38
MEAT AND FISH						
Bovine meat	9.30	2.14	1.3	16.20	3.72	2.46
Chicken meat	2.70	0.45	0.18	2.80	0.6	0.2
Pork meat	4.86	1.03	0.52	5.50	2.25	1.43
Fish: cod	0.67	0.09	0.03	0.60	0.11	0.03
Fish: salmon	13.40	1.88	0.49	15.00	1.64	0.33
Fish: tuna	0.49	0.11	0.04	0.70	0.13	0.05
DAIRY						
Whole milk	3.25	0.86	0.31	3.90	0.98	0.43
Cheese: mozzarella	22.10	5.33	2.44	20.30	5.82	2.03
Cheese: cheddar	33.80	8.7	3.55	34.90	8.68	3.6
OTHERS						
Whole-grain wheat flour	2.50	0.41	0.02	2.00	0.26	0.01
Oats	6.90	1.03	0.06			
Chocolate 70–85% cacao	42.60	10.1	13.6			

^§^ Values expressed as g/100 g food.

**Table 3 ijms-23-06030-t003:** Some fish oil composition found in current supplements on the market (data extracted from ref. [71]).

FAME *	Sample 1 ^¶^	Sample 2 ^¶^	Sample 3 ^¶^	Sample 4 ^¶^	Sample 5 ^¶^	Sample 6 ^¶^
SFA	21.92 ± 0.05	3.66 ± 0.10	32.00 ± 0.17	23.52 ± 0.14	26.28 ± 0.21	31.49 ± 0.13
MUFA	19.49 ± 0.20	17.27 ± 0.23	52.60 ± 0.14	16.34 ± 0.06	1.03 ± 0.04	26.18 ± 0.03
PUFA	55.89 ± 0.25	77.72 ± 0.23	11.51 ± 0.22	57.61 ± 0.05	59.53 ± 0.19	41.22 ± 0.11
PUFA ω-6	4.58 ± 0.20	5.93 ± 0.16	2.12 ± 0.05	4.15 ± 0.09	12.39 ± 0.14	3.29 ± 0.10
PUFA ω-3	52.01 ± 0.31	72.48 ± 0.51	9.38 ± 0.18	54.01 ± 0.06	59.06 ± 0.18	37.93 ± 0.15
total TFA	2.00 ± 0.07	0.66 ± 0.04	3.89 ± 0.10	1.98 ± 0.15	1.25 ± 0.02	1.12 ± 0.13
TFA ω-3	1.40 ± 0.10	0.27 ±0.01	2.43 ± 0.09	1.30 ± 0.14	1.13 ± 0.03	0.27 ± 0.08

* Fatty acid methyl ester (FAME) contents (% rel.); ^¶^ Samples are the interior of capsules from different supplements (1–6) marketed in Spain and Italy. Analyses of the same capsule content performed in triplicates (*n* = 3); TFA ω-3 correspond to mono-trans isomers.

**Table 4 ijms-23-06030-t004:** Lipid peroxidation biomarkers and main reactivity with biomolecules, together with detection and molecular effects in cancer.

Biomarker	Reactivity	Detection	Molecular Effects (Dose-Dependent)	Ref
MDA	Formation of DNA adducts	Serum, urine (high levels)	Carcinogenic Cancer progression Tumor growth inhibitor Apoptotic	[120,121,122]
4-HNE	Protein adducts	Serum, urine (low, medium, high levels)	Carcinogenic Cancer progression Cytotoxic, apoptotic	[123,124,125]
Acrolein	DNA and protein adducts	Blood and urine (high levels)	Neuronal cell death Carcinogenic Cytotoxic	[126,127,128]
Isoprostanes	Stable final product	Urine, plasma, tissue (high levels)	Cell membrane impairment	[129,130,131,132]

## Data Availability

Not applicable.

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
