# Peer review of "Critical Review on Fatty Acid-Based Food and Nutraceuticals as Supporting Therapy in Cancer"

_ijms, 2022, doi:10.3390/ijms23116030_

Round 1
Reviewer 1 Report
The topic of the article is very interesting and important. However, the authors provide only some general information on the topic, without molecular insights. Major parts of the article should be revised and rewritten to be more molecular-oriented. Some suggestions:
- FA chain size and properties and function
- role of FA in the modulation of inflammation in cancer,
- molecular targets for FA,
- modulation of the immune system by FA and its role in cancer.
I would like to suggest some reorganization of the paper, as in many places authors refer to further paragraphs, which impact the clarity of the article.
Part of the “he Lipogenesis and Ketogenic Diets in Cancer” paragraph should be rewritten to be more informative. Especially the part regarding the ketogenic diet should be fixed.
136 – KD diet doesn’t necessarily reduce glucose availability.
130-141 – description of the role of KD in cancer is rather superficial, and some absolute results should be provided when citing original articles.
143 – instead of “fasting blood sugar” should be “fasting blood glucose”
155-157 – the statement is not supported by any data. “successful fatty acid-based cancer therapy” is not defined, there is even no fatty acid-based cancer therapy, and no data supporting that “increased formation of palmitic and stearic acids” can be a measure of the outcome of such a therapy.
168 – “circulating blood fat levels” – authors should be more specific. The authors of the original article assess the influence of specific fatty acids.
204-205 – Authors should consider if Table 2 citing is necessary. Provided FA contents can vary significantly between different parts of the world, and the American diet is ‘highly specific’ and not representative of most countries. Probably, it will be better if the authors provide some region-specific data for comparison, or better some actual data on FA consumption.
271 – “Fatty Acid-Based Membrane Balance for Signaling” paragraph is interesting but presents only basic facts and general knowledge on this topic. Some more details are needed, and a possible mechanism of action needs to be explained. E.g. the role of inflammation in cancer, different receptors for PUFA and FA derivatives, and so on.
346-356 – In one sentence authors stated “In the nutrition facts label of marketed products total lipids are reported, and information of the specific fatty acid types are provided”, but in another “This is an unclear information for consumers missing exactly the quality and quantity of fatty acids”. To me both sentences are contradictory.
455 – “Fatty Acids Peroxidation as Molecular Reactivity in Cancer Therapy” – again very general, what about antioxidant vitamins counteracting pro-oxidative properties of unsaturated fatty acids?
Author Response
Answers to Reviewer # 1
The topic of the article is very interesting and important. However, the authors provide only some general information on the topic, without molecular insights. Major parts of the article should be revised and rewritten to be more molecular-oriented. Some suggestions:
- FA chain size and properties and function
- role of FA in the modulation of inflammation in cancer,
- molecular targets for FA,
- modulation of the immune system by FA and its role in cancer.
I would like to suggest some reorganization of the paper, as in many places authors refer to further paragraphs, which impact the clarity of the article.
answer:
We used a “functional” approach in order to contextualize the data available in the literature on fatty acid-based foods and nutraceuticals in cancer therapy. In particular, we offer molecular insights regarding the role of fatty acid from nutra-strategies in cancer for the control of membrane properties and membrane-based signaling, including the “remodeling” mechanism, that is not frequently considered to explain the importance of the fatty acid-based strategies. We thank and accept the suggestion of this referee and modified several parts of our review in order to introduce the points that have been raised and they can be checked in the “track changes” version of the manuscript:
- we added a sentence in the Abstract in order to clarify that in the review we highlighted the role of membrane fatty acids, also for the application of membrane lipid therapy.
- in the first section a new Figure 1 explains the fatty acid chain size and metabolism in order to explain better the interplay between fatty acids provided by foods and fatty acids obtained from metabolism. For the properties of fatty acids, we added two reviews (new refs11,12) to the already cited refs (12-13 in the old version).
- in section 2 we added the role of FA in modulation of inflammation, especially mentioning the omega-6 and omega-3 intakes with two new references (new refs 19,20)
- we deepened the part regarding the fate of the lipid mediators from fatty acids and their targets in cancer. In particular, to complete also the scenario of the prostaglandin functions, we made some examples on the interaction between PG from arachidonic acid and from di-homo gamma-linolenic acid, adding new refs (refs 62-64) and anticipated the already cited reference of Jara-Gutierrez et al from section 6 to section 4 for deepening the subject treated at this point. We think that the addition of the new references and better explanation in section 6 is satisfactory for providing readers with additional molecular insights, as asked by this referee (see also the answer to another point below).
Part of the “Lipogenesis and Ketogenic Diets in Cancer” paragraph should be rewritten to be more informative. Especially the part regarding the ketogenic diet should be fixed.
answer:
We followed the indication of this referee to ameliorate this section.
136 – KD diet doesn’t necessarily reduce glucose availability.
answer:
We modified the sentence accordingly.
130-141 – description of the role of KD in cancer is rather superficial, and some absolute results should be provided when citing original articles.
answer:
We cited 3 new articles regarding this point, in particular enriching the debate of the results that is ongoing in human studies (refs 39, 43,44)
143 – instead of “fasting blood sugar” should be “fasting blood glucose”
answer:
corrected
155-157 – the statement is not supported by any data. “successful fatty acid-based cancer therapy” is not defined, there is even no fatty acid-based cancer therapy, and no data supporting that “increased formation of palmitic and stearic acids” can be a measure of the outcome of such a therapy.
answer:
Indeed, the referee is right: the meaning of the paragraph was completely different, saying that the increased formation of SFA induces activation of SCD enzymes, that become the target of fatty acid-based strategy. We amended this paragraph.
168 – “circulating blood fat levels” – authors should be more specific. The authors of the original article assess the influence of specific fatty acids.
answer:
We specified fatty acids as requested.
204-205 – Authors should consider if Table 2 citing is necessary. Provided FA contents can vary significantly between different parts of the world, and the American diet is ‘highly specific’ and not representative of most countries. Probably, it will be better if the authors provide some region-specific data for comparison, or better some actual data on FA consumption.
answer:
We evaluated the concern of the referee and, also, we maintain our intention to offer the reader with a list of foods in order to raise concern on this subject. Therefore, we decided to report two different data bases (USA and English ones) and added a sentence to explain the differences in fatty acid composition that can occur depending on the country of origin.
271 – “Fatty Acid-Based Membrane Balance for Signaling” paragraph is interesting but presents only basic facts and general knowledge on this topic. Some more details are needed, and a possible mechanism of action needs to be explained. E.g. the role of inflammation in cancer, different receptors for PUFA and FA derivatives, and so on.
answer:
In this section the target is to indicate the importance of membrane balance regarding fatty acids as precursor of signaling molecules. We mentioned already the role of lipid mediators coming from different fatty acids. To accept also the point of this referee, we made examples of PGE2 and PGE1 molecular mechanisms, but we also addressed readers to the recent and exhaustive review of Jara-Gutiérrez, Á.; Baladrón, V. to deepen this aspect, that was already cited for Section 6, and we anticipated to Section 4. Three new references were added (refs 62-64).
346-356 – In one sentence authors stated “In the nutrition facts label of marketed products total lipids are reported, and information of the specific fatty acid types are provided”, but in another “This is an unclear information for consumers missing exactly the quality and quantity of fatty acids”. To me both sentences are contradictory.
answer:
We amended the paragraphs in order to make them clear and not contradictory.
455 – “Fatty Acids Peroxidation as Molecular Reactivity in Cancer Therapy” – again very general, what about antioxidant vitamins counteracting pro-oxidative properties of unsaturated fatty acids?
answer:
We completed the section following this suggestion and added new references regarding the effects of antioxidant vitamins (new refs 150-155 and ref 157).
Reviewer 2 Report
This manuscript discusses several aspects of fatty acids-based foods and nutraceuticals in cancer therapies. The subject is of interest to many readers, but mainly those interested in the field of cancer as well as that of nutrition. The manuscript is very well written and pleasant to read. I have only four minor points which can be easily corrected by the authors.
1) Abstract (line 18): Define PUFA;
2) Line 282: Define RBC;
3) Line 344: ... in quality and quantity to fulfill ....;
4) Table 3: Specify the meaning of the numbers 1 to 6? (We don't want to go to reference 60 to understand).
Author Response
This manuscript discusses several aspects of fatty acids-based foods and nutraceuticals in cancer therapies. The subject is of interest to many readers, but mainly those interested in the field of cancer as well as that of nutrition. The manuscript is very well written and pleasant to read. I have only four minor points which can be easily corrected by the authors.
answer:
Thank you very much for your appreciation.
- Abstract (line 18): Define PUFA;
Done
- Line 282: Define RBC;
done
- Line 344: ... in quality and quantity to fulfill ....;
done
- Table 3: Specify the meaning of the numbers 1 to 6? (We don't want to go to reference 60 to understand).
done
